# Text2NKG: Fine-Grained N-ary Relation Extraction for N-ary relational Knowledge Graph Construction

## Abstract

Beyond traditional binary relational facts, n-ary relational knowledge graphs (NKGs) are comprised of n-ary relational facts containing more than two entities, which are closer to real-world facts with broader applications. However, the construction of NKGs still significantly relies on manual labor, and n-ary relation extraction still remains at a course-grained level, which is always in a single schema and fixed arity of entities. To address these restrictions, we propose Text2NKG, a novel fine-grained n-ary relation extraction framework for n-ary relational knowledge graph construction. We introduce a span-tuple classification approach with hetero-ordered merging to accomplish fine-grained n-ary relation extraction in different arity. Furthermore, Text2NKG supports four typical NKG schemas: hyper-relational schema, event-based schema, role-based schema, and hypergraph-based schema, with high flexibility and practicality. Experimental results demonstrate that Text2NKG outperforms the previous state-of-the-art model by nearly 20% points in the $F_1$ scores on the fine-grained n-ary relation extraction benchmark in the hyper-relational schema. Our code and datasets are publicly available[1].

## 1 Introduction

Modern knowledge graphs, such as Freebase Bollacker et al. (2008), Google Knowledge Vault Dong et al. (2014), and Wikidata Vrandečić & Krötzsch (2014), convert unstructured knowledge to structured multi-relational graphs with various applications in question-and-answer Yih et al. (2015), query-answering Arakelyan et al. (2021), logical reasoning Chen et al. (2022), and recommendation systems Zhang et al. (2016). Traditional knowledge graphs consist of triple-based facts ($subject$, $relation$, $object$) with two entities and one relation between them Bordes et al.

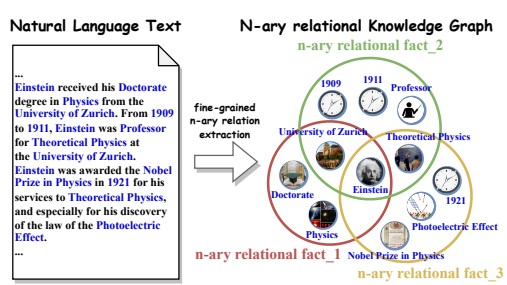

Figure 1: An example of NKG construction.

(2013); Balazevic et al. (2019). However, real-world facts tend to contain more than two entities, which are called n-ary relational facts ($n \geq 2$), which cannot be represented by merging binary relations. For example, consider the statement: "Einstein received his Bachelor degree in Mathematics and his Doctorate degree in Physics." When broken down into binary relations, the facts become: (Einstein, degree, Doctorate degree), (Einstein, major, Physics), (Einstein, degree, Bachelor), and (Einstein, major, Mathematics). However, we can't merge these binary relations effectively because we can't determine whether Einstein's doctoral major was in Physics or Mathematics. This necessitates the use of N-ary relational Knowledge Graphs (NKG) to represent such information, like (Einstein, degree, Doctorate, major, Physics). As shown in Figure 1,

---

[1] Anonymous Github Code: `https://anonymous.4open.science/r/Text2NKG`

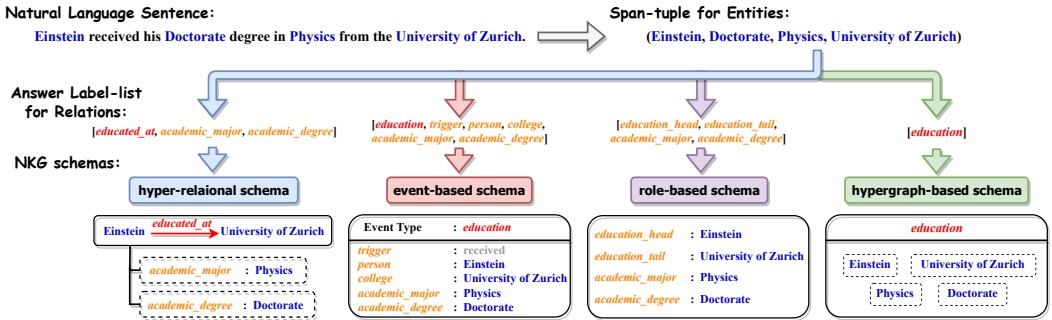

Figure 2: An example of fine-grained n-ary relation extraction in four NKG schemas.

an NKG consists of numerous n-ary relational facts with richer knowledge representation and a wider application capability.

Each NKG has a schema to represent the structure of every n-ary relational fact in the NKG. For example, Wikidata utilizes n-ary relational facts with hyper-relational schema Rosso et al. (2020); Galkin et al. (2020); Wang et al. (2021a), i.e., $(s, r, o, \{(k_i, v_i)\}_{i=1}^{n-2})$ adds $(n-2)$ key-value pairs to the main triple to represent auxiliary information, forming an n-ary relational fact with n entities. In addition to the hyper-relational schema, the n-ary relational facts of NKGs also have event-based schema $(r, \{(k_i, v_i)\}_{i=1}^{n})$ Guan et al. (2022); Lu et al. (2021), role-based schema $(\{(k_i, v_i)\}_{i=1}^{n})$ Guan et al. (2019); Liu et al. (2021) and hypergraph-based schema $(r, \{v_i\}_{i=1}^{n})$ Wen et al. (2016); Fatemi et al. (2021) for different scenarios. Extracting these n-ary relational facts from textual knowledge is called n-ary relation extraction, which is the key step in NKG construction. Taking a real-world textual fact "Einstein received his Doctorate degree in Physics from the University of Zurich." as an example, through n-ary relation extraction, we can extract a four-arity structured span-tuple for entities (Einstein, University of Zurich, Doctorate, Physics) with an answer label-list for relations accordingly as a 4-ary relational fact from the sentence as shown in Figure 2.

However, most existing NKGs, such as JF17K Wen et al. (2016), Wikipeople Guan et al. (2019), WD50K Galkin et al. (2020), EventKG Guan et al. (2022), etc., are constructed manually but not automatically. The key step of knowledge graph construction is relation extraction, but most relation extraction methods target traditional binary relational facts Wang & Lu (2020); Zhong & Chen (2021); Ye et al. (2022). The n-ary relation extraction methods are currently always focused on the course-grained extraction with solid keys Jia et al. (2019); Zhuang et al. (2022), but are not competent for fine-grained NKG construction with various n-ary relations. Recently, CubeRE Chia et al. (2022) proposes the cube-filling method, the only fine-grained n-ary relation extraction method. Nevertheless, it can only perform hyper-relational extraction with limited accuracy and cannot cover other useful NKG schemas.

To address these challenges, we propose a novel n-ary relation extraction framework, Text2NKG, which automates the generation of n-ary relational facts from natural language text for NKG construction. Text2NKG proposes a span-tuple multi-label classification method with hetero merging, which converts n-ary relation extraction into a multi-label classification problem for span-tuples consisting of all arrangements of three entities in a sentence. The number of labels is determined by the number of relations in the selected NKG schema. Text2NKG can be applied to all NKG schemas, with hyper-relational schema, event-based schema, role-based schema, and hypergraph-based schema provided as examples, which have a wide range of applications.

In addition, we extend the current n-ary relation extraction benchmark HyperRED Chia et al. (2022), which is only in the hyper-relational schema, to four NKG schemas. We've done sufficient n-ary relation extraction experiments on HyperRED, and the experimental results show that Text2NKG achieves nearly 20 percentage points ahead of the existing state-of-the-art model CubeRE in $F_1$ scores of hyper-relational extraction. We also compared the results of Text2NKG in four schemas to verify applications. We are excited to open-source our complete code and are willing to contribute to the knowledge graph construction community.

## 2 RELATED WORK

**N-ary relational Knowledge Graph.** An n-ary relational knowledge graph (NKG) consists of n-ary relational facts, which contain $n$ entities ($n \geq 2$) and several relations. The n-ary relational facts are necessary and cannot be replaced by combinations of some binary relational facts because we cannot distinguish which binary relations are combined to represent the n-ary relational fact in the whole KG. Therefore, NKG utilizes a schema in every n-ary relational fact locally and a hypergraph representation globally. Firstly, the simplest NKG schema is hypergraph-based. Wen et al. (2016) found that over 30% of Freebase Bollacker et al. (2008) entities participate facts with more than two entities, first defined n-ary relations mathematically and used star-to-clique conversion to convert triple-based facts representing n-ary relational facts into the first NKG dataset JF17K in hypergraph-based schema $(r, \{v_i\}_{i=1}^n)$. Fatemi et al. (2021) proposed FB-AUTO and M-FB15K with the same hypergraph-based schema. Secondly, Guan et al. (2019) introduced role information for n-ary relational facts and extracted Wikipeople, the first NKG dataset in role-based schema $(\{(k_i, v_i)\}_{i=1}^n)$, composed of role-value pairs. Thirdly, Wikidata Vrandečić & Krötzsch (2014), the largest knowledge base, utilizes an NKG schema based on hyper-relation $(s, r, o, \{(k_i, v_i)\}_{i=1}^{n-2})$, which adds auxiliary key-value pairs to the main triple. Galkin et al. (2020) first proposed an NKG dataset in hyper-relational schema WD50K. Fourthly, as Guan et al. (2022) pointed out, events are also n-ary relational facts. One basic event representation has an event type, a trigger, and several key-value pairs Lu et al. (2021). Regarding the event type as the main relation, the (trigger: value) as one of the key-value pairs, and the arguments as the rest key-value pairs, we can obtain an event-based NKG schema $(r, \{(k_i, v_i)\}_{i=1}^n)$.

Based on four common NKG schemas, we propose Text2NKG, the first method for extraction of structured n-ary relational facts from natural language text, which improves NKG representation and application.

**N-ary Relation Extraction.** Relation extraction is an important part of knowledge graph construction, directly affecting the quality, scale, and application of KGs. While most of the current n-ary relation extraction for NKG construction depends on manual construction Wen et al. (2016); Guan et al. (2019); Galkin et al. (2020) but not automated methods. Most automated relation extraction methods target the extraction of traditional binary relational facts. For example, Wang & Lu (2020) proposes a table-filling method for binary relation extraction, and Zhong & Chen (2021); Ye et al. (2022) propose span-based relation extraction methods with levitated marker and packed levitated marker, respectively. For n-ary relation automated extraction, some approaches Jia et al. (2019); Jain et al. (2020); Viswanathan et al. (2021) treat n-ary relation extraction as a binary classification problem and predict whether the composition of n-ary information in a document is valid or not. However, these methods extract n-ary information in fixed arity, which are not flexible. Moreover, most of these methods are based on the course-grained level with solid keys, which is not competent for fine-grained NKG construction with various n-ary relations. Recently, Chia et al. (2022) proposes the only automated n-ary relation extraction method, CubeRE, which extends the table-filling extraction method to n-ary relation extraction with cube-filling. However, it can only model hyper-relational schema with limited extraction accuracy.

In this paper, we propose the first fine-grained n-ary relation extraction framework Text2NKG for NKG construction in four example schemas, proposing a span-tuple multi-label classification method with hetero-ordered merging to improve the accuracy of hyper-relational extraction substantially.

## 3 PRELIMINARIES

**Formulation of NKG.** An NKG $\mathcal{G} = \{\mathcal{E}, \mathcal{R}, \mathcal{F}\}$ consists of an entity set $\mathcal{E}$, a relation set $\mathcal{R}$, and an n-ary fact (n≥2) set $\mathcal{F}$. Each n-ary fact $f^n \in \mathcal{F}$ consists of entities $\in \mathcal{E}$ and relations $\in \mathcal{R}$. For hyper-relational schema Rosso et al. (2020): $f_{hr}^n = (e_1, r_1, e_2, \{r_{i-1}, e_i\}_{i=3}^n)$ where $\{e_i\}_{i=1}^n \in \mathcal{E}$, $\{r_i\}_{i=1}^{n-1} \in \mathcal{R}$. For event-based schema Lu et al. (2021): $f_{ev}^n = (r_1, \{r_{i+1}, e_i\}_{i=1}^n)$, where $\{e_i\}_{i=1}^n \in \mathcal{E}$, $\{r_i\}_{i=1}^{n+1} \in \mathcal{R}$. For role-based schema Guan et al. (2019): $f_{ro}^n = (\{r_i, e_i\}_{i=1}^n)$, where $\{e_i\}_{i=1}^n \in \mathcal{E}$, $\{r_i\}_{i=1}^n \in \mathcal{R}$. For hypergraph-based schema Wen et al. (2016): $f_{hg}^n = (r_1, \{e_i\}_{i=1}^n)$, where $\{e_i\}_{i=1}^n \in \mathcal{E}$, $r_1 \in \mathcal{R}$.

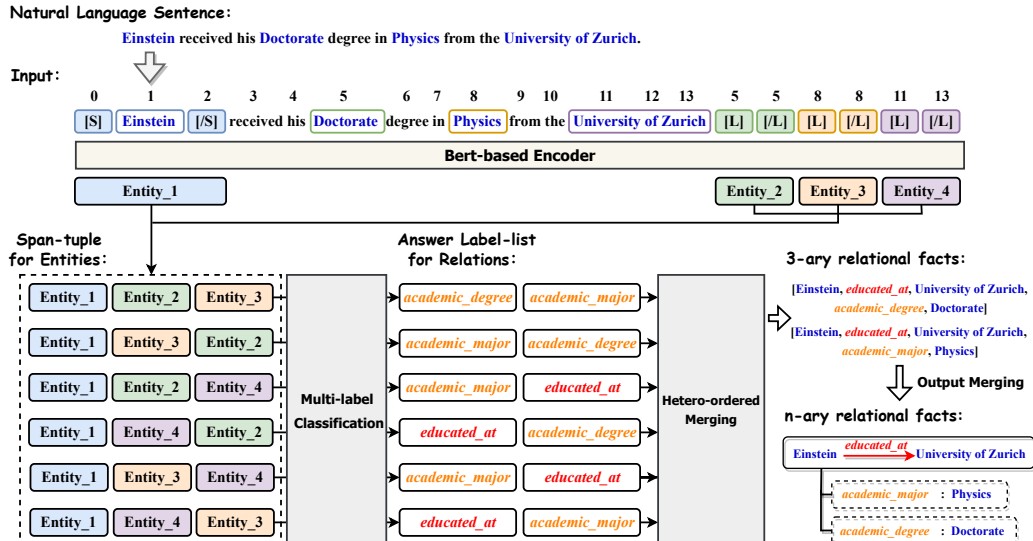

Figure 3: An overview of Text2NKG extracting n-ary relation facts from a natural language sentence in hyper-relational NKG schema for an example.

**Problem Definition.** Given an input sentence with $l$ words $s = \{w_1, w_2, ..., w_l\}$, an entity $e$ is a consecutive span of words: $e = \{w_p, w_{p+1}, ..., w_q\} \in \mathcal{E}_s$, where $p, q \in \{1, ..., l\}$, and $\mathcal{E}_s = \{e_j\}_{j=1}^m$ is the entity set of all $m$ entities in the sentence. The output of n-ary relation extraction, $R()$, is a set of n-ary relational facts $\mathcal{F}_s$ in given NKG schema in $\{f_{hr}^n, f_{ev}^n, f_{ro}^n, f_{hg}^n\}$. Specifically, each n-ary relational fact $f^n \in \mathcal{F}_s$ is extracted by multi-label classification of one of the ordered span-tuple for $n$ entities $[e_i]_{i=1}^n \in \mathcal{E}_s$, forming an answer label-list for $n_r$ relations $[r_i]_{i=1}^{n_r} \in \mathcal{R}$, where $n$ is the arity of the extracted n-ary relational fact, and $n_r$ is the number of answer relations in the fact, which is determined by the given NKG schema: $R([e_i]_{i=1}^n) = [r_i]_{i=1}^{n-1}$, when $f^n = f_{hr}^n$, $R([e_i]_{i=1}^n) = [r_i]_{i=1}^{n+1}$ when $f^n = f_{ev}^n$, $R([e_i]_{i=1}^n) = [r_i]_{i=1}^n$ when $f^n = f_{ro}^n$, and $R([e_i]_{i=1}^n) = [r_1]$ when $f^n = f_{hg}^n$.

## 4 METHODOLOGY

In this section, we first introduce the overview of the Text2NKG framework, followed by the span-tuple multi-label classification, training strategy, hetero-ordered merging, and output merging.

### 4.1 OVERVIEW OF TEXT2NKG

Text2NKG is a fine-grained n-ary relation extraction framework built for n-ary relational knowledge graph (NKG) construction. The input to Text2NKG is natural language text tokens labeled with entity span in sentence units. We **extract 3-ary facts as an atomic unit** and then **merge them into n-ary facts** later to realize n-ary extraction of arbitrary arity. Because if using binary facts, merging them into n-ary facts based on shared elements within these facts will lead to misunderstandings as analyzed in Section 1. On the other hand, using facts with four entities or more makes it challenging to judge which of the included 3-ary facts can be extracted as independent facts.

Specifically, inspired by Ye et al. (2022), Text2NKG first encodes the entities using BERT-based Encoder Devlin et al. (2019) with a packaged levitated marker for embedding. Then, each arrangement of ordered span-tuple with three entity embeddings will be classified with multiple labels, and the framework will be learned by the weighted cross-entropy with a null-label bias. In the decoding stage, in order to filter the n-ary relational facts whose entity compositions have isomorphic hetero-ordered characteristics, Text2NKG proposes a hetero-ordered merging strategy to merge the label probabilities of $3! = 6$ arrangement cases of span-tuples composed of the same entities and filter out the output 3-ary relational facts existing non-conforming relations. Finally, Text2NKG combines the output 3-ary relational facts to form the final n-ary relational facts with output merging.

## 4.2 SPAN-TUPLE MULTI-LABEL CLASSIFICATION

For the given sentence token $s = \{w_1, w_2, ..., w_l\}$ and the set of entities $\mathcal{E}_s$, in order to perform fine-grained n-ary relation extraction, we need first to encode a span-tuple $(e_1, e_2, e_3)$ consisting of every arrangement of three ordered entities, where $e_1, e_2, e_3 \in \mathcal{E}_s$. Due to the high time complexity of training every span-tuple as one training item, inspired by Ye et al. (2022), we achieve the reduction of training items by using packed levitated markers that pack one training item with each entity in $\mathcal{E}_s$ separately. Specifically, in each packed training item, a pair of solid tokens, [S] and [/S], are added before and after the packed entity $e_S = \{w_{p_S}, ..., w_{q_S}\}$, and $(|\mathcal{E}_s| - 1)$ pairs of levitated markers, [L] and [/L], according to other entities in $\mathcal{E}_s$, are added with the same position embeddings as the beginning and end of their corresponding entities span $e_{L_i} = \{w_{p_{L_i}}, ..., w_{q_{L_i}}\}$ to form the input token $\mathbf{X}$:

$$
\begin{aligned}
\mathbf{X} = \{ & w_1, ..., [S], w_{p_S}, ..., w_{q_S}, [/S], ..., \\
& w_{p_{L_i}} \cup [L], ..., w_{q_{L_i}} \cup [/L], ..., w_l \}.
\end{aligned}
\tag{1}
$$

We encode such token by the BERT-based pre-trained model encoder Devlin et al. (2019):

$$
\{h_1, h_2, ..., h_t\} = \text{BERT}(\mathbf{X}),
\tag{2}
$$

where $t = |\mathbf{X}|$ is the imput token length, $\{h_i\}_{i=1}^{t} \in \mathbb{R}^d$, and $d$ is embedding size.

There are several span-tuples $(A, B, C)$ in a training item. The embedding of first entity $h_A \in \mathbb{R}^{2d}$ in the span-tuple is obtained by concat embedding of the solid markers, [S] and [/S], and the embeddings of second and third entities $h_B, h_C \in \mathbb{R}^{2d}$ are obtained by concat embeddings of levitated markers, [L] and [/L] with all $A_{m-1}^2$ arrangement of any other two entities in $\mathcal{E}_s$. Thus, we obtain the embedding representation of the three entities to form $A_{m-1}^2$ span-tuples in one training item. Therefore, every input sentence contains $m$ training items with $m A_{m-1}^2 = A_m^3$ span-tuples for any ordered arrangement of three entities.

We then define $3 \times n_r$ linear classifiers $\{\text{FNN}_i^k\}_{i=1}^{n_r}, k = 1, 2, 3$ to classify the span-tuples for multiple-label classification. Each classifier targets the prediction of one relation $r_i$, thus obtaining a probability lists $(\mathbf{P}_i)_{i=1}^{n_r}$ with all relations in given relation set $\mathcal{R}$ plus a null-label:

$$
\mathbf{P}_i = \text{FNN}_i^1(h_A) + \text{FNN}_i^2(h_B) + \text{FNN}_i^3(h_C),
\tag{3}
$$

where $\text{FNN}_i^k \in \mathbb{R}^{2d \times (|\mathcal{R}|+1)}$, and $\mathbf{P}_i \in \mathbb{R}^{(|\mathcal{R}|+1)}$.

## 4.3 TRAINING STRATEGY

In order to train the $n_r$ classifiers for each relation prediction more accurately, we performed data augmentation strategy in terms of span-tuples. Taking the hyper-relational schema as an example, given a hyper-relational fact $(A, r_1, B, r_2, C)$, we consider swapping the head and tail entities and changing the main relation to the corresponding inverse relation $(B, r_1^{-1}, A, r_2, C)$, as well as swapping the tail entities and auxiliary values, and swapping the main relation and the auxiliary key $(A, r_2, C, r_1, B)$ also as labeled training span-tuple cases. Thus $R_{hr}(A, B, C) = (r_1, r_2)$ can be augmented with $3! = 6$ orders of span-tuples:

$$
\begin{cases}
R_{hr}(A, B, C) = (r_1, r_2), \\
R_{hr}(B, A, C) = (r_1^{-1}, r_2), \\
R_{hr}(A, C, B) = (r_2, r_1), \\
R_{hr}(B, C, A) = (r_2, r_1^{-1}), \\
R_{hr}(C, A, B) = (r_2^{-1}, r_1), \\
R_{hr}(C, B, A) = (r_1, r_2^{-1}).
\end{cases}
\tag{4}
$$

For other schemas, we can also obtain 6 fully-arranged cases of labeled span-tuples in a similar way, as described in Appendix A. If no n-ary relational fact exists between the three entities of span-tuples, then relation labels are set as null-label.

Since most cases of span-tuple are null-label, we set a weight hyperparameter $\alpha \in (0, 1]$ between the null-labels and other labels to balance the learning of the null-label. We jointly trained the $n_r$

| Dataset | #Ent | #R_hr | #R_ev | #R_ro | #R_hg | All | | Train | | Dev | | Test | |
|---|---|---|---|---|---|---|---|---|---|---|---|---|---|
| | | | | | | #Sentence | #Fact | #Sentence | #Fact | #Sentence | #Fact | #Sentence | #Fact |
| HyperRED | 40,293 | 106 | 232 | 168 | 62 | 44,840 | 45,994 | 39,840 | 39,978 | 1,000 | 1,220 | 4,000 | 4,796 |

Table 1: Dataset statistics, where the columns indicate the number of entities, relations with four schema, sentences and n-ary relational facts in all sets, train set, dev set, and test set,respectively.

classifiers for each relations by cross-entropy loss $\mathcal{L}$ with a null-label weight bias $\mathbf{W}_\alpha$:

$$\mathcal{L} = -\sum_{i=1}^{n_r} \mathbf{W}_\alpha \log \left( \frac{\exp(\mathbf{P}_i[r_i])}{\sum_{j=1}^{|\mathcal{R}|+1} \exp(\mathbf{P}_{ij})} \right), \tag{5}$$

where $\mathbf{W}_\alpha = [\alpha, 1.0, 1.0, ...1.0] \in \mathbb{R}^{(|\mathcal{R}|+1)}$.

## 4.4 HETERO-ORDERED MERGING

In the decoding stage, since Text2NKG labels all 6 different arrangement of the same entity composition, we design a hetero-ordered merging strategy to merge the corresponding labels of these 6 hetero-ordered span-tuples into one to generate non-repetitive n-ary relational facts unsupervisedly. For hyper-relational schema ($n_r = 2$), we combine the predicted probabilities of two labels $\mathbf{P}_1, \mathbf{P}_2$ in 6 orders to $(A, B, C)$ order as follows:

$$\begin{cases} \mathbf{P}_1 = \mathbf{P}_1^{(ABC)} + I(\mathbf{P}_1^{(BAC)}) + \mathbf{P}_2^{(ACB)} \\ \quad + I(\mathbf{P}_2^{(BCA)}) + \mathbf{P}_2^{(CAB)} + \mathbf{P}_1^{(CBA)}, \\ \mathbf{P}_2 = \mathbf{P}_2^{(ABC)} + \mathbf{P}_2^{(BAC)} + \mathbf{P}_1^{(ACB)} \\ \quad + \mathbf{P}_1^{(BCA)} + I(\mathbf{P}_1^{(CAB)}) + I(\mathbf{P}_2^{(CBA)}), \end{cases} \tag{6}$$

where $I()$ is a function for swapping the predicted probability of relations and the corresponding inverse relations. Then, we take the maximum probability to obtain labels $r_1, r_2$, forming a 3-ary relational fact $(A, r_1, B, r_2, C)$ and filter it out if there are null-labels in $(r_1, r_2)$. If there are inverse relation labels in $(r_1, r_2)$, we can also transform the order of entities and relations as equation 4. For event-based schema, role-based schema, and hypergraph-based schema, all can be generated by hetero-ordered merging according to this idea, shown in Appendix B.

## 4.5 OUTPUT MERGING

After hetero-ordered merging, we merge the output 3-ary relational facts to form higher-arity facts, with hyper-relational schema based on the same main triple, event-based schema based on the same main relation (event type), role-based schema based on the same key-value pairs, and hypergraph-based schema based on the same hyperedge relation. This way, we can **unsupervisedly** obtain n-ary relational facts **with dynamic number of arity numbers** for NKG construction.

## 5 EXPERIMENTS

This section presents the experimental setup, results, and analysis. We answer the following research questions (RQs): **RQ1**: Does Text2NKG outperform other fine-grained n-ary relation extraction methods? **RQ2**: Whether Text2NKG can cover NKG construction for various schemas? **RQ3**: Does the main components of Text2NKG work? **RQ4**: How does the null-label bias hyperparameter in Text2NKG affect performance? **RQ5**: Can Text2NKG get complete n-ary relational facts in different arity? **RQ6**: How does Text2NKG perform in specific case study? **RQ7**: What is the future development of Text2NKG in the era of large language models?

## 5.1 EXPERIMENTAL SETUP

**Datasets.** The HyperRED Chia et al. (2022) dataset is the only existing dataset for extracting n-ary relations with annotated extracted entities. Therefore, we expand the HyperRED dataset to four

| Model | PLM | HyperRED : Hyper-relational / Dev | | | HyperRED : Hyper-relational / Test | | |
|---|---|---|---|---|---|---|---|
| | | Precision | Recall | $F_1$ | Precision | Recall | $F_1$ |
| **Unsupervised Method** | | | | | | | |
| ChatGPT | gpt-3.5-turbo ($\approx$175B) | 12.0583 | 11.2764 | 11.6542 | 11.4021 | 10.9134 | 11.1524 |
| GPT-4 | gpt-4 ($\approx$1760B) | 15.7324 | 15.2377 | 15.4811 | 15.8187 | 15.4824 | 15.6487 |
| **Supervised Method** | | | | | | | |
| Generative Baseline | | 63.79 ± 0.27 | 59.94 ± 0.68 | 61.80 ± 0.37 | 64.60 ± 0.47 | 59.67 ± 0.35 | 62.03 ± 0.21 |
| Pipelinge Baseline | | 69.23 ± 0.30 | 58.21 ± 0.57 | 63.24 ± 0.44 | 69.00 ± 0.48 | 57.55 ± 0.19 | 62.75 ± 0.29 |
| CubeRE | | 66.14 ± 0.88 | 64.39 ± 1.23 | 65.23 ± 0.82 | 65.82 ± 0.84 | 64.28 ± 0.25 | 65.04 ± 0.29 |
| Text2NKG w/o DA | BERT-base (110M) | 76.02 ± 0.50 | 72.28 ± 0.68 | 74.10 ± 0.55 | 73.55 ± 0.81 | 70.63 ± 1.40 | 72.06 ± 0.34 |
| Text2NKG w/o $\alpha$ | | 88.77 ± 0.85 | 78.39 ± 0.47 | 83.26 ± 0.70 | 88.09 ± 0.69 | 76.64 ± 0.45 | 81.97 ± 0.58 |
| Text2NKG w/o HM | | 61.74 ± 0.34 | 76.97 ± 0.44 | 68.52 ± 0.69 | 61.07 ± 0.73 | 76.16 ± 0.59 | 67.72 ± 0.48 |
| Text2NKG (ours) | | **91.26 ± 0.69** | **79.36 ± 0.51** | **84.89 ± 0.44** | **90.77 ± 0.60** | **77.53 ± 0.32** | **83.63 ± 0.63** |
| Generative Baseline | | 67.08 ± 0.49 | 65.73 ± 0.78 | 66.40 ± 0.47 | 67.17 ± 0.40 | 64.56 ± 0.58 | 65.84 ± 0.25 |
| Pipelinge Baseline | BERT-large (340M) | 70.58 ± 0.78 | 66.58 ± 0.66 | 68.52 ± 0.72 | 69.21 ± 0.55 | 64.27 ± 0.24 | 66.65 ± 0.28 |
| CubeRE | | 68.75 ± 0.82 | 68.88 ± 1.03 | 68.81 ± 0.46 | 66.39 ± 0.96 | 67.12 ± 0.69 | 66.75 ± 0.28 |
| Text2NKG (ours) | | **91.90 ± 0.79** | **79.43 ± 0.42** | **85.21 ± 0.69** | **91.06 ± 0.81** | **77.64 ± 0.46** | **83.81 ± 0.54** |

Table 2: Comparison of Text2NKG with other baselines in the hyper-relational extraction on HyperRED. Results of the supervised baseline models are mainly taken from the original paper Chia et al. (2022). The best results in each metric are in **bold**.

| Model | PLM | HyperRED : Event-based | | | HyperRED : Role-based | | | HyperRED : Hypergraph-based | | |
|---|---|---|---|---|---|---|---|---|---|---|
| | | Precision | Recall | $F_1$ | Precision | Recall | $F_1$ | Precision | Recall | $F_1$ |
| **Unsupervised Method** | | | | | | | | | | |
| ChatGPT | gpt-3.5-turbo ($\approx$175B) | 10.4678 | 11.1628 | 10.8041 | 11.4387 | 10.4203 | 10.9058 | 11.2998 | 11.7852 | 11.5373 |
| GPT-4 | gpt-4 ($\approx$1760B) | 13.3681 | 14.6701 | 13.9888 | 13.6397 | 12.5355 | 13.0643 | 13.0907 | 13.6701 | 13.3741 |
| **Supervised Method** | | | | | | | | | | |
| Text2Event | | 73.94 | 70.56 | 72.21 | 72.73 | 68.45 | 70.52 | 73.68 | 70.37 | 71.98 |
| UIE | T5-base (220M) | 76.51 | 73.02 | 74.72 | 72.17 | 69.84 | 70.98 | 72.03 | 68.74 | 70.34 |
| LasUIE | | 79.62 | 78.04 | 78.82 | 77.01 | 74.26 | 75.61 | 76.21 | 73.75 | 74.96 |
| Text2NKG (ours) | BERT-base (110M) | **86.20** | **79.25** | **82.58** | **86.72** | **78.94** | **82.64** | **83.53** | **86.59** | **85.03** |
| Text2Event | | 75.58 | 72.39 | 73.97 | 73.21 | 70.85 | 72.01 | 75.28 | 72.73 | 73.98 |
| UIE | T5-large (770M) | 79.38 | 74.69 | 76.96 | 74.47 | 71.84 | 73.14 | 74.57 | 71.93 | 73.22 |
| LasUIE | | 81.29 | 79.54 | 80.40 | 79.37 | 76.63 | 77.97 | 77.49 | 74.96 | 76.20 |
| Text2NKG (ours) | BERT-large (340M) | **88.47** | **80.30** | **84.19** | **86.87** | **80.86** | **83.76** | **85.06** | **86.72** | **85.89** |

Table 3: Comparison of Text2NKG with other baselines for n-ary relation extraction in event-based, role-based, and hypergraph-based schemas on HyperRED. The best results in each metric are in **bold**.

schemas as standard fine-grained n-ary relation extraction benchmarks and conduct experiments on them. The statistics of the HyperRED with four schemas are shown in Table 1, and the construction detail is in Appendix C.

**Baselines.** We compare Text2NKG against **Generative Baseline** Lewis et al. (2020), **Pipeline Baseline** Wang et al. (2021b), and **CubeRE** Chia et al. (2022) in fine-grained n-ary relation extraction task of hyper-relational schema. For n-ary relation extraction in the other three schemas, we compared Text2NKG with event extraction models such as **Text2Event** Lu et al. (2021), **UIE** Lu et al. (2022), and **LasUIE** Fei et al. (2022). Furthermore, we utilized different prompts to test the currently most advanced large-scale pre-trained language models **ChatGPT** Wei et al. (2023) and **GPT-4** OpenAI (2023) in an unsupervised manner, specifically for the extraction performance across the four schemas. The detailed baseline settings can be found in Appendix D.

**Ablations.** To evaluate the significance of Text2NKG's three main components, data augmentation (DA), null-label weight hyperparameter ($\alpha$), and hetero-ordered merging (HM), we obtain three simplified model variants by removing any one component from the model (**Text2NKG w/o DA**, **Text2NKG w/o** $\alpha$, and **Text2NKG w/o HM**) for comparison.

**Evaluation Metrics.** We use the $F_1$ score with precision and recall to evaluate the dev set and the test set. For a predicted n-ary relational fact to be considered correct, the entire fact must match the ground facts completely.

**Hyperparameters and Enviroment.** We train 10 epochs on HyperRED using the Adam optimizer. All experiments were done on a single NVIDIA A100 GPU, and all experimental results were derived by averaging five random seed experiments. Appendix E shows Text2NKG's optimal hyperparameter settings. Appendix F shows training details.

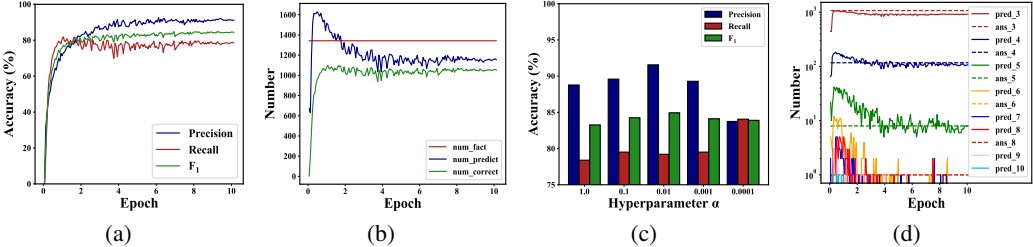

Figure 4: (a) Precision, Recall, and $F_1$ changes in the dev set during the training of Text2NKG. (b) The changes of the number of true facts, the number of predicted facts, and the number of predicted accurate facts during the training of Text2NKG. (c) Precision, Recall, and $F_1$ results on different null-label hyperparameter ($\alpha$) settings. (d) The changes of the number of extracted n-ary relation extraction in different arity.

## 5.2 MAIN RESULTS (RQ1)

The experimental results of proposed Text2NKG and other baselines with both BERT-base and BERT-large encoders can be found in Table 2 for the fine-grained n-ary relation extraction in hyper-relational schema. We can observe that Text2NKG shows a huge improvement over the existing optimal model CubeRE on both the dev and test datasets of HyperRED. The $F_1$ score is improved by 19.66 percentage points in the dev set and 18.60 percentage points in the test set with the same BERT-base encoder, and 16.40 percentage points in the dev set and 17.06 percentage points in the test set with the same BERT-large encoder, reflecting Text2NKG's excellent performance. Figure 4(a) and 4(b) intuitively show the changes of evaluation metrics and answers of facts in the dev set during the training of Text2NKG. It is worth noting that Text2NKG exceeds 90% in precision accuracy, which proves that the model can obtain very accurate n-ary relational facts and provides a good guarantee for the quality of fine-grained NKG construction.

## 5.3 RESULTS ON VARIOUS NKG SCHEMAS (RQ2)

As shown in Table 3, besides hyper-relational schema, Text2NKG also accomplishes the tasks of fine-grained n-ary relation extraction in three other different NKG schemas on HyperRED, which demonstrates good utility. In the added tasks of n-ary relation extraction for event-based, role-based, and hypergraph-based schemas, since no model has done similar experiments at present, we used event extraction or unified extraction methods such as Text2Event Lu et al. (2021), UIE Lu et al. (2022), and LasUIE Fei et al. (2022) for comparison. We found that Text2NKG still works best in these schemas, which demonstrates good versatility.

## 5.4 ABLATION STUDY (RQ3)

Data augmentation (DA), null-label weight hyperparameter ($\alpha$), and hetero-ordered merging (HM) are the three main components of Text2NKG. For the different Text2NKG variants as shown in Table 2, it can be observed that DA, $\alpha$, and HM all contribute to the accurate results of our complete model. By comparing the differences, we find that HM is most effective by combining the probabilities of labels of different orders, followed by DA and $\alpha$.

## 5.5 ANALYSIS OF NULL-LABEL WEIGHT HYPERPARAMETERS (RQ4)

We compared the effect for different null-label weight hyperparameters ($\alpha$). As shown in Figure 4(c), the larger the $\alpha$, the greater the learning weight of null-label compared with other lables, the more relations are predicted as null-label. After filtering out the facts having null-label, fewer facts are extracted, so the precision is generally higher, and the recall is generally lower. The smaller the $\alpha$, the more relations are predicted as non-null labels, thus extracting more n-ary relation facts, so the recall is generally higher, and the precision is generally lower. Comparing the results of $F_1$ values for different $\alpha$, it is found that $\alpha = 0.01$ works best. When applied in practice, the hyperparameter $\alpha$ can be adjusted according to specific needs to obtain the best results.

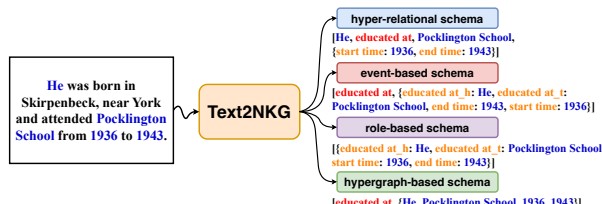

Figure 5: Case study of Text2NKG's n-ary relation extraction with four schemas on HyperRED.

### 5.6 ANALYSIS OF N-ARY RELATION EXTRACTION IN DIFFERENT ARITY (RQ5)

We use output merging to address the dynamic changes in the number of elements in n-ary relational facts. For instance, in the hyper-relational fact (`Einstein, educated_at, University of Zurich, degree: Doctorate degree, major: Physics`), the Text2NKG algorithm allows us to extract two 3-ary atomic facts: (`Einstein, educated_at, University of Zurich, degree: Doctorate degree`) and (`Einstein, educated_at, University of Zurich, major: Physics`). These are then merged based on the same primary triple (`Einstein, educated_at, University of Zurich`) to form a 4-ary fact. The same principle applies to facts of higher arities.

Figure 4(d) shows the number of n-ary relational facts extracted after output merging and the number of the answer facts in different arity during training of Text2NKG on the dev set. We find that, as the training proceeds, the final output of Text2NKG converges to the correct answer in terms of the number of complete n-ary relational facts in each arity, achieving implementation of n-ary relation extraction in indefinite arity unsupervised, with good scalability.

### 5.7 CASE STUDY (RQ6)

Figure 5 shows a case study of n-ary relation extraction by a trained Text2NKG. For a natural language sentence, "`He was born in Skirpenbeck, near York and attended Pocklin.`", four structured n-ary relation extraction can be obtained by Text2NKG according to the requirements. Taking the hyper-relational schema for an example, Text2NKG can successfully extract one n-ary relational fact consisting of a main triple [`He, educated at, Pocklington`], and two auxiliary key-value pairs {`start time:1936`}, {`end time:1943`}. This intuitively validates the practical performance of Text2NKG on the fine-grained n-ary relation extraction to better contribute to the NKG construction.

### 5.8 COMPARISON WITH CHATGPT AND GPT-4 (RQ7)

As shown in Table 2 and Table 3, we compared the extraction effects under four NKG schemas of the supervised Text2NKG with the unsupervised ChatGPT and GPT-4. We found that, these large language models cannot accurately distinguish the closely related relations in the fine-grained NKG relation repository, resulting in their F1 scores ranging around 10%-15%, which is much lower than the performance of Text2NKG. On the other hand, the limitation of Text2NKG is that its performance is confined within the realm of supervised training. Therefore, in future improvements and practical applications, we suggest combining small supervised models with large unsupervised models to better balance solving the cold-start and fine-grained accuracy problems in NKG construction. Appendix G.

## 6 CONCLUSION

In this paper, we propose Text2NKG, a novel fine-grained n-ary relation extraction framework for n-ary relational knowledge graph (NKG) construction. Experimental results show that Text2NKG outperforms other baselines on fine-grained n-ary relation extraction tasks, with nearly 20 percentage points improvement in $F_1$ scores. Moreover, Text2NKG supports n-ary relation extraction in four schemas: hyper-relational, event-based, role-based, and hypergraph-based. Meanwhile, we extend HyperRED dataset to a fine-grained n-ary relation extraction benchmark in four schemas.

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

## APPENDIX

## A SUPPLEMENT TO DATA AUGMENTATION

In addition to the hyper-relational schema, the data augmentation strategies for other schemas are as follows:

For event-based schema, given an event-based fact $(r_1, r_2, A, r_3, B, r_4, C)$, we consider keeping the main relation $r_1$ unchanged, and swapping other key-value pairs, $\{r_2, A\}$, $\{r_3, B\}$, and $\{r_4, C\}$, positionally, also as labeled training span-tuple cases. Thus $R_{ev}(A, B, C) = (r_1, r_2, r_3, r_4)$ can be augmented with 6 orders of span-tuples:

$$
\begin{cases}
R_{ev}(A, B, C) = (r_1, r_2, r_3, r_4), \\
R_{ev}(B, A, C) = (r_1, r_3, r_2, r_4), \\
R_{ev}(A, C, B) = (r_1, r_2, r_4, r_3), \\
R_{ev}(B, C, A) = (r_1, r_3, r_4, r_2), \\
R_{ev}(C, A, B) = (r_1, r_4, r_2, r_3), \\
R_{ev}(C, B, A) = (r_1, r_4, r_3, r_2).
\end{cases}
\tag{7}
$$

For role-based schema, given a role-based fact $(r_1, A, r_2, B, r_3, C)$, we consider swapping key-value pairs, $\{r_1, A\}$, $\{r_2, B\}$, and $\{r_3, C\}$, positionally, also as labeled training span-tuple cases. Thus $R_{ro}(A, B, C) = (r_1, r_2, r_3)$ can be augmented with 6 orders of span-tuples:

$$
\begin{cases}
R_{ro}(A, B, C) = (r_1, r_2, r_3), \\
R_{ro}(B, A, C) = (r_2, r_1, r_3), \\
R_{ro}(A, C, B) = (r_1, r_3, r_2), \\
R_{ro}(B, C, A) = (r_2, r_3, r_1), \\
R_{ro}(C, A, B) = (r_3, r_1, r_2), \\
R_{ro}(C, B, A) = (r_3, r_2, r_1).
\end{cases}
\tag{8}
$$

For hypergraph-based schema, given a hypergraph-based fact $(r_1, A, B, C)$, we consider keeping the main relation $r_1$ unchanged, and swapping entities, $A$, $B$, and $C$, positionally, also as labeled training span-tuple cases. Thus $R_{hg}(A, B, C) = (r_1)$ can be augmented with 6 orders of span-tuples:

$$
\begin{cases}
R_{hg}(A, B, C) = (r_1), \\
R_{hg}(B, A, C) = (r_1), \\
R_{hg}(A, C, B) = (r_1), \\
R_{hg}(B, C, A) = (r_1), \\
R_{hg}(C, A, B) = (r_1), \\
R_{hg}(C, B, A) = (r_1).
\end{cases}
\tag{9}
$$

## B SUPPLEMENT TO HETERO-ORDERED MERGING

In addition to the hyper-relational schema, the hetero-ordered merging strategies for other schemas are as follows:

For event-based schema ($n_r = 4$), we combine the predicted probabilities of four labels $\mathbf{P}_1, \mathbf{P}_2, \mathbf{P}_3, \mathbf{P}_4$ in 6 orders to $(A, B, C)$ order as follows:

$$
\begin{cases}
\mathbf{P}_1 = \mathbf{P}_1^{(ABC)} + \mathbf{P}_1^{(BAC)} + \mathbf{P}_1^{(ACB)} \\
\qquad + \mathbf{P}_1^{(BCA)} + \mathbf{P}_1^{(CAB)} + \mathbf{P}_1^{(CBA)}, \\
\mathbf{P}_2 = \mathbf{P}_2^{(ABC)} + \mathbf{P}_3^{(BAC)} + \mathbf{P}_2^{(ACB)} \\
\qquad + \mathbf{P}_4^{(BCA)} + \mathbf{P}_3^{(CAB)} + \mathbf{P}_4^{(CBA)}, \\
\mathbf{P}_3 = \mathbf{P}_3^{(ABC)} + \mathbf{P}_2^{(BAC)} + \mathbf{P}_4^{(ACB)} \\
\qquad + \mathbf{P}_2^{(BCA)} + \mathbf{P}_4^{(CAB)} + \mathbf{P}_3^{(CBA)}, \\
\mathbf{P}_4 = \mathbf{P}_4^{(ABC)} + \mathbf{P}_4^{(BAC)} + \mathbf{P}_3^{(ACB)} \\
\qquad + \mathbf{P}_3^{(BCA)} + \mathbf{P}_2^{(CAB)} + \mathbf{P}_2^{(CBA)}.
\end{cases}
\tag{10}
$$

Then, we take the maximum probability to obtain labels $r_1, r_2, r_3, r_4$, forming a 3-ary relational fact $(r_1, r_2, A, r_3, B, r_4, C)$ and filter it out if there are null-labels in $(r_1, r_2, r_3, r_4)$.

For role-based schema ($n_r = 3$), we combine the predicted probabilities of three labels $\mathbf{P}_1, \mathbf{P}_2, \mathbf{P}_3$ in 6 orders to $(A, B, C)$ order as follows:

$$
\begin{cases}
\mathbf{P}_1 = \mathbf{P}_1^{(ABC)} + \mathbf{P}_2^{(BAC)} + \mathbf{P}_1^{(ACB)} \\
\qquad + \mathbf{P}_3^{(BCA)} + \mathbf{P}_2^{(CAB)} + \mathbf{P}_3^{(CBA)}, \\
\mathbf{P}_2 = \mathbf{P}_2^{(ABC)} + \mathbf{P}_1^{(BAC)} + \mathbf{P}_3^{(ACB)} \\
\qquad + \mathbf{P}_1^{(BCA)} + \mathbf{P}_3^{(CAB)} + \mathbf{P}_2^{(CBA)}, \\
\mathbf{P}_3 = \mathbf{P}_3^{(ABC)} + \mathbf{P}_3^{(BAC)} + \mathbf{P}_2^{(ACB)} \\
\qquad + \mathbf{P}_2^{(BCA)} + \mathbf{P}_1^{(CAB)} + \mathbf{P}_1^{(CBA)}.
\end{cases}
\tag{11}
$$

Then, we take the maximum probability to obtain labels $r_1, r_2, r_3$, forming a 3-ary relational fact $(r_1, A, r_2, B, r_3, C)$ and filter it out if there are null-labels in $(r_1, r_2, r_3)$.

For hypergraph-based schema ($n_r = 1$), we combine the predicted probabilities of one label $\mathbf{P}_1$ in 6 orders to $(A, B, C)$ order as follows:

$$
\begin{cases}
\mathbf{P}_1 = \mathbf{P}_1^{(ABC)} + \mathbf{P}_1^{(BAC)} + \mathbf{P}_1^{(ACB)} \\
\qquad + \mathbf{P}_1^{(BCA)} + \mathbf{P}_1^{(CAB)} + \mathbf{P}_1^{(CBA)}.
\end{cases}
\tag{12}
$$

Then, we take the maximum probability to obtain labels $r_1$, forming a 3-ary relational fact $(r_1, A, B, C)$ and filter it out if $r_1$ is null-label.

## C  CONSTRUCTION OF DATASET

Based on the original hyper-relational schema on HyperRED dataset Chia et al. (2022), we construct other three schemas (event-based, role-based, and hypergraph-based) for fine-grained n-ary relation extraction. Firstly, we view the main relation in the hyper-relational schema as the event type in the event-based schema, combine the head entity and tail entity with two extra head key and tail key to convert them into two key-value pairs, and remain the auxiliary key-value pairs in the hyper-relational schema. Taking 'Einstein received his Doctorate degree in Physics from the University of Zurich.' as an example, it can be represented as (Einstein, educated, University of Zurich, {academic_major, Physics}, {academic_ degree, Doctorate}) in the hyper-relational schema and (education, {trigger, received}, {person, Einstein}, {college, University of Zurich}, {academic_major, Physics},{academic_degree, Doctorate}) in the event-based schema. Secondly, we remove the event type in the event-based schema to obtain the role-based schema. Thirdly, we remove all the keys in key-value pairs and remain the relation to build the hypergraph-based schema.

# D    BASELINE SETTINGS

Firstly, for the original hyper-relational schema of HyperRED, we adopted the same baselines as in the CubeRE paper Chia et al. (2022) to compare with Text2NKG:

**Generative Baseline:**    Generative Baseline uses BART Lewis et al. (2020), a sequence-to-sequence model, to transform input sentences into a structured text sequence.

**Pipeline Baseline:**    Pipeline Baseline uses UniRE Wang et al. (2021b) to extract relation triplets in the first stage and a span extraction model based on BERT-Tagger Devlin et al. (2019) to extract value entities and corresponding qualifier labels in the second stage.

**CubeRE:**    CubeRE Chia et al. (2022) is the only hyper-relational extraction model that uses a cube-filling model inspired by table-filling approaches and explicitly considers the interaction between relation triplets and qualifiers.

Secondly, for the event-based schema, role-based schema, and hypergraph-based schema, we added the following baselines to further validate the effect of Text2NKG on the fine-grained N-ary relation fact extraction task in the HyperRED dataset:

**Text2Event:**    Text2Event Chia et al. (2022) is a classic model in the Event extraction domain. However, it is not applicable to extractions of the hyper-relational schema. For the role-based schema extraction, we retained the key without referring to the main relation, while for the hypergraph-based schema extraction, we retained the main relation without referring to the key to get the final result for comparison.

**UIE / LasUIE:**    UIE Lu et al. (2022) and LasUIE Fei et al. (2022) are unified information extraction models that can handle most tasks like NER, RE, EE, etc. However, they are still only suitable for event extraction in the multi-relational extraction domain and are not applicable to extractions of the hyper-relational schema. Therefore, we adopted the same approach as with Text2Event to compare with Text2NKG.

Thirdly, under the impact of the wave of large-scale language models brought about by ChatGPT on traditional natural language processing tasks, we added unsupervised large models as baselines to compare with Text2NKG in the n-ary relation extraction tasks of the four schemas.

**ChatGPT / GPT4:**    Using different prompts, we tested the latest state-of-the-art large-scale pre-trained language models ChatGPT Wei et al. (2023) and GPT-4 OpenAI (2023) in an unsupervised manner, evaluating their performance on the extraction of the four schemas.

# E    HYPERPARAMETER SETTINGS

We use the grid search method to select the optimal hyperparameter settings for both Text2NKG with Bert-base and Bert-large. We use the same hyperparameter settings in Text2NKG with different encoders. The hyperparameters that we can adjust and the possible values of the hyperparameters are first determined according to the structure of our model in Table 4. Afterward, the optimal hyperparameters are shown in **bold**.

| Hyperparameter | HyperRED |
|:---:|:---:|
| $\alpha$ | $\{1.0, 0.1, \mathbf{0.01}, 0.001\}$ |
| Train_batch_size | $\{2, 4, \mathbf{8}, 16\}$ |
| Eval_batch_size | $\{\mathbf{1}\}$ |
| Learning rate | $\{1e-5, \mathbf{2e\text{-}5}, 5e-5\}$ |
| Max_sequence_length | $\{128, \mathbf{256}, 512, 1024\}$ |
| Weight decay | $\{\mathbf{0.0}, 0.1, 0.2, 0.3\}$ |

Table 4:  Hyperparameter Selection.

## F    MODEL TRAINING DETAILS

We train 10 epochs on HyperRED with the optimal combination of hyperparameters. Text2NKG and all its variants have been trained on a single NVIDIA A100 GPU. Using our optimal hyperparameter settings, the time required to complete the training on HyperRED is 4h with BERT-base encoder and 10h with BERT-large encoder.

## G    DISCUSSION OF ADVANCED LLMS SUCH AS CHATGPT IN FINE-GRAINED N-ARY RELATION EXTRACTION TASKS

We have tried to use LLM APIs such as ChatGPT and GPT to do similar n-ary relation extraction tasks, i.e., prompting model input and output formats for extraction. The advantage of ChatGPT is that it can perform similar tasks in a few-shot situation, however, for building high-quality knowledge graphs, the performance and the fineness of the n-ary relation extraction are much lower than Text2NKG. This is because ChatGPT is not good at multi-label classification tasks that contain less semantic interpretation. When the number of labels of relations in our relation collection is very large, we need to write a very long prompt to tell the LLM about our label candidate collection, which again leads to the problem of forgetting. Therefore, we have tried numerous prompt templates to enhance the extraction effect of ChatGPT, however, on fine-grained n-ary relation extraction task, the best result of ChatGPT can only reach about 10% of $F_1$ value on HyperRED, which is much lower than the result of 80%+ $F_1$ value of Text2NKG.

However, advanced LLMs such as ChatGPT are a good idea for training dataset generation for Text2NKG in such tasks to save some manual labor to only verify and correct the training items generated. For future work, we will continue our research in this direction and try to combine large language models with Text2NKG-like supervised models for automated fine-grained n-ary relation extraction for n-ary relational knowledge graph construction.

