# OpenReview forum: "Text2NKG: Fine-Grained N-ary Relation Extraction for N-ary relational Knowledge Graph Construction"
_ICLR.cc/2024/Conference — ICLR 2024 Conference Withdrawn Submission_

### Official Review · Reviewer_JaVm · 2023-10-14

**Soundness:** 3 good
**Presentation:** 2 fair
**Contribution:** 3 good
**Rating:** 5
**Confidence:** 3

**Summary:**

The paper introduces Text2NKG, a novel framework for fine-grained n-ary relation extraction in the construction of n-ary relational knowledge graphs (NKGs). It addresses the limitations of manual labor in NKG construction and the coarse-grained nature of n-ary relation extraction. Text2NKG employs a span-tuple classification method with hetero-ordered merging to extract n-ary relations with high precision. It supports various NKG schemas, including hyper-relational, event-based, role-based, and hypergraph-based schemas. Experimental results demonstrate that Text2NKG surpasses previous state-of-the-art models by nearly 20% in F1 scores, particularly in the hyper-relational schema.

Contributions:
1. Text2NKG: The paper proposes a novel framework, which automates the extraction of n-ary relational facts from natural language text to construct NKGs.

2. Fine-grained n-ary relation extraction: Text2NKG introduces a span-tuple multi-label classification method with hetero merging, enabling precise extraction of n-ary relations with different arities.

3. Support for diverse NKG schemas: Text2NKG accommodates four typical NKG schemas, enhancing flexibility and practicality in knowledge graph construction.

**Strengths:**

Apart from dataset and benchmarking, Strengths including:

1. Text2NKG offers a significant advancement in automating the extraction of n-ary relational facts from natural language text. This automation reduces the reliance on manual labor, making the construction of n-ary relational knowledge graphs (NKGs) more efficient and scalable.

2. The framework introduces a fine-grained approach to n-ary relation extraction. By employing a span-tuple multi-label classification method with hetero merging, Text2NKG can handle n-ary relations of different arities. This precision in extraction is crucial for capturing nuanced relationships in complex datasets.

3. Text2NKG stands out in its versatility, as it supports four typical NKG schemas, including hyper-relational, event-based, role-based, and hypergraph-based schemas. This flexibility enables the framework to be applied across a wide range of domains and use cases, making it highly adaptable to various real-world applications.


The paper is well-written and easy to follow and The examples given in the article can help to better understand the paper.
It embodies the spirit of open source and contribution.

The development of Text2NKG has broader implications for the field of knowledge representation and reasoning.

**Weaknesses:**

Weaknesses:

1. While the extension of the HyperRED dataset is a valuable contribution, the paper could benefit from additional experimentation on a wider range of datasets, especially in diverse domains. This would further validate the framework's performance and applicability across various knowledge graph construction scenarios.

2. The paper does not provide a detailed discussion for the computational resources required to implement Text2NKG. This information is crucial for practitioners and researchers to understand the feasibility and potential resource constraints associated with adopting the framework.

3. The paper does not extensively discuss potential overfitting issues. Since fine-grained relation extraction involves complex pattern recognition, it is essential to address how Text2NKG generalizes to new, unseen data, especially in real-world, noisy text environments.

4. While Text2NKG demonstrates substantial improvements over the previous state-of-the-art model, CubeRE, a more comprehensive comparison with a wider range of alternative approaches would provide a more complete assessment of its relative strengths and weaknesses.

5. The paper does not explicitly address potential biases that may arise during the relation extraction process. Natural language data can contain biases that may be inadvertently learned by the model, which could impact the quality of the extracted relations.

6. The technical details are not described clearly and somehow seems lack of novelty.

Overall, the workload of this paper is not particularly large, and the complexity is not that high.
The experiments and diagrams mentioned in the article are not so sufficient. More experimental materials can be provided to form a strong support for the experimental results.
The article lacks discussion on the limitations of the model and could briefly describe its potential defects and areas for future improvement.
An overview of Text2NKG described in Figure 3 should be a very important diagram for the article, but the architecture of the entire model seems to be not regular enough, and there are not many annotations in the diagram, such as solid tokens "[S] and [/S]" for entity delimitation and annotations for levitated markers "[L] and [/L]".
In METHODOLOGY, there is a lack of detailed description of the working principle of the linear classifier FNN. In addition, the introduction in the OUTPUT MERGING part is too general. It is best to describe the novel method proposed in this article in detail so that readers can better understand it.
In the ablation study, three important components are suddenly mentioned, including data augmentation (DA), null-label weight hyperparameter (α), and hetero-ordered merging (HM). However, the previous METHODOLOGY chapter does not seem to emphasize the importance of the first two components.
The HyperRED dataset is the only existing dataset for extracting nary relations with annotated extracted entities. How can the generalization ability of the model be demonstrated?
The structure in INTRODUCTION is slightly confusing, such as whether the second paragraph can be considered in the following chapter, and then whether the third paragraph should appear in related work.
Some of the content in the article is duplicated, resulting in the compression of the substantive content. For example, in RELATED WORK "An n-ary relational knowledge graph (NKG) consists of n-ary relational facts, which contain n entities (n ≥ 2)...... because we cannot distinguish which binary relations are combined to represent the n-ary relational fact in the whole KG. "is repeated in the INTRODUCTION, and some places can be pointed out briefly.
There are some minor errors in the grammar and format of the paper, such as "course-grained" should be "coarse-grained", and "imput" in "where t = | X | is the imput token length". In addition, it is best to add parentheses to the citations in the text, otherwise the content will look confusing and difficult to read.

**Questions:**

1. Could you provide more details on the computational resources required to implement Text2NKG?

2. While the extension of the HyperRED dataset is valuable, have you considered experimenting with a wider range of datasets in diverse domains to further validate the performance and applicability of Text2NKG?

3. How does Text2NKG generalize to new, unseen data, especially in noisy, real-world text environments?

4. Could you explain a bit more about the scalability of Text2NKG when dealing with large corpora of natural language text? How does the framework handle processing extensive datasets, and have you encountered any computational challenges in scaling up?

5. In the context of relation extraction, have you considered potential biases in the natural language data that Text2NKG may encounter? How do you address or mitigate any biases that may be inadvertently learned by the model during the extraction process?

6. In terms of the performance of fine grains, can more experimental results be provided to confirm?

7. After span-tuple multi-label classification, how is the answer label-list for relations obtained in so many classifications?

8.Is only one training item depicted in Figure 3? If so, it is best to explain it so as not to create the misunderstanding that the subject is to be fixed. Additionally, an explanation for the variable 'm' in SPAN-TUPLE MULTI-LABEL CLASSIFICATION can be provided.

9.Why does the data augmentation strategy mentioned in TRAINING STRATEGY result in more accurate training after conducting swapping and inverse operations? What is the underlying mechanism?

10. In the Eq. 2, what is the main reason to choose BERT? Can we use other models instead?

---

### Official Review · Reviewer_z8Bh · 2023-10-30

**Soundness:** 3 good
**Presentation:** 2 fair
**Contribution:** 2 fair
**Rating:** 5
**Confidence:** 4

**Summary:**

This paper proposes Text2NKG, a framework for n-ary relation extraction that infers structured relational facts from raw text. The key contribution is a novel procedure to extract n-ary relational tuples representing semantic connections between multiple entity mentions in a sentence.

**Strengths:**

1. The authors demonstrate the effectiveness of their proposed framework on the HyperRED benchmark. Their approach achieves strong performance compared to certain prior methods.
2. Their experiments demonstrate that their framework outperforms some other methods on three versions of HyperRED.
3. The authors provide a clear explanation of their framework and how it works. The methodology is welldescribed and easy to understand.
4. The code implementation for their framework is publicly available, enabling reproducibility of the results. This is a strength of the work.

**Weaknesses:**

1. I believe it should be coarse-grained?
2. The quality of the writing in this paper leaves room for improvement. Normally, the introduction should briefly outline the motivation behind the research and explain the methodology. However, this paper's introduction seems to be excessively brief when it comes to motivation and methodology. Additionally, in related work section, the phrase "we propose..." has been reiterated twice. This excessive emphasis on their framework's function seems pointlessly repetitive.
3. Noticing only two referenced works from 2023 (ChatGPT and GPT-4) raises questions regarding the absence of related papers from the same year. One could question if no other papers published in 2023 bear relevance to their work.
4. The range of experiments conducted appears to be limited, with insufficient diversity in datasets and methods being compared.
5. The paper seems to have a lackluster impact on the specific field it targets.

**Questions:**

You've stated that your framework is specifically designed for fine-grained n-ary relation extraction. Could you provide some clarity on what makes it particularly suitable for this task? Couldn't it also be applicable for coarse-grained n-ary relation extraction?

---

### Official Review · Reviewer_ZHPg · 2023-10-30

**Soundness:** 2 fair
**Presentation:** 2 fair
**Contribution:** 2 fair
**Rating:** 3
**Confidence:** 4

**Summary:**

To address issues that the challenge of constructing n-ary relational knowledge graphs (NKGs), this paper proposes a new method termed Text2NKG, which has been a labor-intensive task due to coarse-grained n-ary relation extraction that relies on a single schema and fixed arity of entities. The framework introduces a novel fine-grained n-ary relation extraction method using a span-tuple classification approach combined with hetero-ordered merging, allowing for extraction in different arities and supporting a variety of NKG schemas like hyper-relational, event-based, role-based, and hypergraph-based with enhanced flexibility and practicality.  Moreover, extensive evaluation verifies the effectiveness and superiority of the model.

**Strengths:**

1.	The proposed method Text2NKG to construct n-ary relational knowledge graphs. To the best of my knowledge, this method will be beneficial to the community.
2.	The authors conducted extensive experiments to demonstrate the effectiveness of the method, which presented the performance of different research questions from various perspectives.

**Weaknesses:**

1.	The paper is not organized clearly, which is not friendly for understanding. Specifically, there are lack the more details and essential discussion about extracting 3-ary facts as an atomic unit.
2.	The algorithms in this paper are not explained in detail, which may cause the fusion for readers. Moreover, some variable symbols in algorithms are not interpreted such as the TRAINING STRATEGY.

**Questions:**

Please refer to the weaknesses.

---

### Official Review · Reviewer_vB44 · 2023-10-31

**Soundness:** 3 good
**Presentation:** 1 poor
**Contribution:** 2 fair
**Rating:** 3
**Confidence:** 3

**Summary:**

The paper introduces a novel approach for extracting n-ary facts that go beyond the traditional triplet format <h, r, t>. In this n-ary setting, facts can encompass related entities (e.g., time, location, etc.) and their associated relationships within a single tuple. To extract such facts, the model begins by extracting a fundamental 3-ary fact. The input sentence is encoded by packed levitated markers [Ye et al. (2022)] followed by a BERT encoder. Following this, three separate linear transformations are applied to the entity encoding for relation decision. Subsequently, these 3-ary relational facts can potentially be merged to form higher-arity facts. Experimental results on the HyperRED dataset demonstrate the performance of the proposed model compared to two basic baselines and a dedicated model CubeRE.

Although the problem of extracting n-ary facts is interesting, the paper's methodological contributions are somewhat limited, especially within the ICLR community. Furthermore, the quality of the representation provided in the paper is suboptimal, with some critical technical details left unclear. For example, the process of merging 3-ary facts into higher-order facts lacks clarity. As a result, I recommend rejecting this paper.

**Strengths:**

As existing models predominately focus on traditional triplet format <h, r, t>, the problem of extracting n-ary facts is interesting.

**Weaknesses:**

1. Technical novelty is limited. The model's approach is rather conventional, involving the extraction of 3-ary facts through existing methods and their subsequent merging to create higher-order models.
2. The paper's representation of the methodology falls short. There are notable deficiencies in explaining crucial technical aspects. E.g., the process of merging 3-ary facts to form higher-order facts lacks clarity.
3. The experimental evaluation is not sufficient. The study only utilizes a single dataset, HyperRED, and does not include comparisons with diverse datasets such as JF17K, FB-AUTO, and M-FB15K.

**Questions:**

1. The text in Figure 1 is too tiny to read.
2. The problem definition part (on page 4) is not presented in a clear manner, which can lead to confusion among readers. For instance, there is a confusing use of "R" and "\matchcal R" for different concepts. It's unclear what "R()" represents—whether it's the output of n-ary relation extraction or part of the NKG schema.
3. On page 5, when (A, B, C) represents a span-tuple, it's unclear what "A^2_{m−1}" signifies.
4. It's not evident why Equation 3 is considered a multiple-label classification problem. It appears to resemble a conventional multi-class classification problem.
5. Since the decoding relies on merging 3-ary tuples, it's unclear how the proposed model supports the other four schemas of n-ary facts, such as the event-based schema depicted in Figure 5.